# Extracellular Vesicles Released by Colorectal Cancer Cell Lines Modulate Innate Immune Response in Zebrafish Model: The Possible Role of Human Endogenous Retroviruses

**DOI:** 10.3390/ijms20153669

**Published:** 2019-07-26

**Authors:** Luca Ferrari, Marco Cafora, Federica Rota, Mirjam Hoxha, Simona Iodice, Letizia Tarantini, Maria Dolci, Serena Delbue, Anna Pistocchi, Valentina Bollati

**Affiliations:** 1EPIGET LAB, Department of Clinical Sciences and Community Health, Università degli Studi di Milano, 20122 Milan, Italy; 2Department of Medical Biotechnology and Translational Medicine, Università degli Studi di Milano, 20129 Milan, Italy; 3Department of Biomedical, Surgical and Dental Sciences, Università degli Studi di Milano, 20133 Milan, Italy; 4Fondazione IRCCS Ca’ Granda Ospedale Maggiore Policlinico, Unit of Occupational Medicine, 20122 Milan, Italy

**Keywords:** extracellular vesicles, HERV, innate immune response, zebrafish

## Abstract

Extracellular vesicles (EVs) are important components of the metastatic niche and are crucial in infiltration, metastasis, and immune tolerance processes during tumorigenesis. We hypothesized that human endogenous retroviruses (HERV) positive EVs derived from tumor cellsmay have a role in modulating the innate immune response. The study was conducted in two different colorectal cancer cell lines, representing different stages of cancer development: Caco-2, derived from a non-metastatic colorectal adenocarcinoma, and SK-CO-1, derived from metastatic colorectal adenocarcinoma (ascites). Both cell lines were treated with decitabine to induce global hypomethylation and to reactivate HERV expression. EVs were quantified by nanoparticle tracking analysis, and HERV-positive EV concentrations were measured by flow cytometry. The effect of EVs isolated from both untreated and decitabine-treated cells on the innate immune response was evaluated by injecting them in zebrafish embryos and then assessing Interleukin 1β (*IL1-β*), Interleukin 10 (*IL-10*), and the myeloperoxidase (*mpx*) expression levels by real-time qPCR. Interestingly, HERV-K positive EVs concentrations were significantly associated with a reduced expression of *IL1-β* and *mpx*, supporting our hypothesis that HERV-positive EVs may act as immunomodulators in tumor progression. The obtained results open new perspectives about the modulation of the immune response in cancer therapy.

## 1. Introduction

Tumor cells proliferate, infiltrate, and metastasize by forming a metastatic niche that can be defined as a supportive and receptive microenvironment, which evades the surveillance by the immune system [1,2,3]. Malignant cells apply several immune evasion strategies, and emerging data suggest that an important role is exerted by the tumor-induced modulation of the innate immune system [4]. 

The key components in the crosstalk between cancer and the surrounding microenvironment include tumor-derived secreted factors, bone marrow-derived cells, suppressive immune cells, host stromal cells, and extracellular vesicles (EVs) [5]. In particular, EVs have a recognized role in promoting tumor progression, in inducing normal cell transformation, in remodeling the parenchymal tissue, and in modulating the immune system [6,7]. 

Several membrane-bound determinants of tumor-derived EVs are involved in cancer infiltration and in immune tolerance. Among these, a growing interest has been turned to human endogenous retroviruses (HERVs) [8,9,10]. About 8.5% of the human genome consists of *HERV* genes, and they are currently classified according to the first letter of the tRNA amino acid core of the primary binding site, which starts reverse transcription. Their genomic structure shares similarities with known exogenous retroviruses, as it consists of *gag*, *pro*, *pol*, and *env* genes, flanked by two long terminal repeats (LTR) that act as promoters [11,12]. HERVs are involved in several physiological processes, such as trophoblast fusion during placental development [13,14] and modulation of the immune response [15,16]. However, these crucial functions may turn pathological and be implicated in the development of different types of human cancers, such as breast cancer, lung cancer, prostate cancer, testicular cancer, seminoma, hepatocellular carcinoma, colorectal cancer, melanomas, germ cell tumor leukemia, and lymphoma [12,17,18,19,20,21]. The mechanisms underlying this evidence are still unclear, but the transactivation of HERVs is supposed to promote carcinogenesis, either by directly effecting HERV components (i.e., mRNAs or functional proteins), or by indirectly activating tumor-associated genes [22,23]. *HERV* promoters are normally forced into a repressed state by DNA methylation [24,25], but can be switched on (e.g., in response to environmental stimuli or viral infections), eventually acting as triggers of an immune modulatory cascade. Moreover, some HERVs (e.g., HERV-K and HERV-H) have been detected in the blood and tissues of patients with cancer [26,27]. In particular, HERVs are involved in colorectal cancer tumorigenesis [28,29,30,31,32].

In the present study, we investigated the hypothesis that EVs derived from colorectal cancer cell lines are involved in the modulation of the innate immune response, which is a central step in the formation of the metastatic niche. We conducted our experiments in two different colorectal cancer cell lines, which represented two different stages in cancer development: Caco-2, derived from a non-metastatic epithelial colorectal adenocarcinoma; SK-CO-1, derived from a metastatic colorectal adenocarcinoma (ascites). We focused on the role of HERV-positive EVs (i.e., HERV-K, HERV-W) as possible immune-modulators. Therefore, we treated both colorectal cancer cell lines with decitabine, in order to induce a global hypomethylation and reactivate *HERVs* expression. We further evaluated the cellular production of total EVs and HERV-positive EVs and, finally, tested their effect on innate immune response, by injecting them in zebrafish embryos. This model is suitable for studying this mechanism because the adaptive immune system has not yet developed in the first month of life [33]. 

## 2. Results

### 2.1. HERV Methylation Analysis

We quantified the methylation levels of *HERV-H*, *HERV-K*, *HERV-W*, and *HERV-P* LTRs, and observed that the Caco-2 cell line showed significantly higher baseline methylation levels compared to the SK-CO-1 cell line for all the *HERV* genes analyzed, while the *LINE-1* methylations levels, representative of global methylation, were similar for both cell lines (Table 1). In order to increase the *HERVs* expression, we treated the Caco-2 and SK-CO-1 cell lines with the hypomethylating agent decitabine. As expected, the treatment induced a significant hypomethylation of *LINE-1* sequences (Appendix A), both in the Caco-2 (untreated: mean ± standard deviation (SD) = 56.9 ± 3.3; treated: mean ± SD = 33.8 ± 0.6; *p*-value < 0.001) and in the SK-CO-1 (untreated: mean ± SD = 51.5 ± 1.6; treated: mean ± SD = 38.7 ± 2.8; *p*-value < 0.001) cells.

Moreover, decitabine induced a profound hypomethylation of *HERV* LTRs in both cell lines (Figure 1; Table 2). The difference in methylation between treated and untreated cells was significant for all the *HERV* LTRs analyzed, except for the *HERV-H* and *HERV-P* LTRs in the SK-CO-1 cells, although this observation might be due to their extremely low basal methylation level. 

To further characterize the functional effects of hypomethylation due to decitabine treatment, *HERV* (Figure 2) *env* expression levels were evaluated by qPCR in both cell lines. In the Caco-2 cell line, only *HERV-H* expression was significantly higher following decitabine treatment (*p* < 0.001) (Figure 2, Appendix A), but *HERV-K* and *HERV-W* also showed a tendency toward increased expression levels. In the SK-CO-1 cell line, the expression levels of all investigated *HERV* genes were significantly increased between untreated and decitabine-treated cells (Figure 2, Appendix A).

### 2.2. Evaluation and Characterization of EVs

We measured the concentration of EVs isolated from the untreated and decitabine-treated Caco-2 and SK-CO-1 cell lines by nanoparticles tracking analysis (NTA). We analyzed these values, comparing decitabine-treated and untreated samples in terms of the mean release of exosomes (i.e., 30–130 nm EVs), microvesicles (i.e., 130–700 nm EVs), and total EVs (i.e., 30–700 nm EVs) (Figure 3). We observed an increase in total EVs release in the decitabine-treated Caco-2 cell line, with an average of 2.01 × 10^9^/mL (±4.71 × 10^9^/mL), compared to the untreated, with an average of 1.47 × 10^9^/mL (±3.69 × 10^9^/mL) (*p* = 0.05) (Figure 3). 

No significant differences were detected between untreated and decitabine-treated SK-CO-1 cells (Figure 3). We further analyzed the effect of decitabine treatment in terms of the distribution of mean vesicle concentrations for each EV size. In the upper part of each panel of Figure 4A, we reported the mean concentration for all EV sizes (i.e., from 30 to 700 nm). The comparisons among the EV sizes are shown in Figure 4B, where the two *p*-values obtained, comparing decitabine-treated versus untreated (Poisson linear regression models for repeated measures at each size), respectively, for the Caco-2 and SK-CO-1 cells, were reported. EVs derived from Caco-2 cells were generally increased after decitabine treatment; on the contrary, EVs from the SK-CO-1 cells were reduced.

We hypothesized that HERV-positive EVs may be involved in innate immune modulation. Therefore, the samples were further characterized for HERV-K and HERV-W positive EVs by flow cytometry analysis (Table 3). 

Unfortunately, commercial antibodies against HERV-H and HERV-P are not currently available, therefore limiting our ability to explore these HERV-positive EVs subtypes. Since retroviruses share the same size of EVs, we evaluated the possibility of having a retroviral contamination in the EV pellet. Therefore, we measured HERV-W and HERV-K positive events in samples without gating for CFSE staining, and observed that HERV-K and HERV-W concentrations were the same in both cases (i.e., either gating or not for CFSE), as reported in the Appendix A for Caco-2 cells, and Appendix A for SK-CO-1 cells. Considering the subpopulations of EVs analyzed by high-resolution flow cytometry, these results minimize the possibility of pelleting HERV-W and HERV-K positive particles with dimensions ≥ 100 nm.

In both the untreated Caco-2 and SK-CO-1 cells, HERV-K and HERV-W positive EVs subtypes were equally represented. Treatment with decitabine did not determine any significant variation in HERVs-positive EVs concentrations (Appendix A).

We also evaluated the epithelial cell adhesion molecule (EpCAM)-positive EVs, as EpCAM is an epithelial cell marker, and therefore should be present in the total amount of microvesicles derived from these cells. The release of EpCAM-positive EVs was higher for the Caco-2 cells than the SK-CO-1 cells (Table 3), thereby confirming the NTA observation. In addition, we quantified the tetraspanin CD63 positive EVs as a non-tissue specific marker for EVs (Table 3) [34].

### 2.3. Effects on Innate Immune Response in Zebrafish Embryos

EVs derived from both the Caco-2 and SK-CO-1 cell lines were injected in the duct of Cuvier, and therefore into the blood circulation, of embryos at 48 h post fertilization (hpf). Cell mediums Dulbecco’s Modified Eagle’s Medium (DMEM) and Eagle’s Minimum Essential Medium (EMEM), for Caco-2 and SK-CO-1 cells, respectively, were injected as control. For each treatment, 40–45 embryos were injected and the experiment was repeated at least three-fold for each condition: control embryos, embryos injected with the Caco-2 EVs, embryos injected with the SK-CO-1 EVs, and embryos injected with the EVs derived from decitabine-treated Caco-2 and SK-CO-1 cells. Because zebrafish embryos only have innate immunity during the first month of development, we were able to investigate the effect of EVs on this specific immune component, while avoiding the confounding effect of the adaptive immune system. To evaluate whether the EVs isolated from both the Caco-2 and SK-CO-1 untreated cells might induce an immune reaction, we injected EVs and measured the expression of cytokines at 20 h post injection (hpi). Specifically, we measured the pro-inflammatory cytokine *Interleukin 1-β* (*IL1-β*), the anti-inflammatory cytokine *Interleukin* 10 (*IL-10*), and the *myeloperoxidase* (*mpx*) gene, ortholog of the mammalian Mpo, and specific markers of granulocytes (Figure 5). We found that the expression of *IL1-β* was significantly lower in embryos injected with EVs derived from the untreated Caco-2 (*p* = 0.034) and the SK-CO-1 cells (*p* = 0.021), compared to those injected with control medium (Figure 5A,B). *IL1-β* expression levels were also lower following the injection of EVs derived from decitabine-treated Caco-2 cells (*p* = 0.003), while no significant changes were observed for SK-CO-1 cells (Figure 5A,B). Therefore, the results indicate a decrease in pro-inflammatory response following the injection of both Caco-2 and SK-CO-1 EVs into zebrafish embryos. The anti-inflammatory response, measured in terms of *IL-10* expression, was observed in parallel to the strong inactivation of the pro-inflammatory response, as in the case of the injection of EVs derived from decitabine-treated Caco-2 cells (*p* < 0.001), and untreated SK-CO-1-derived EVs (*p* = 0.012) (Figure 5C,D). 

The expression of *mpx*, marking neutrophil activation, was not similar to *IL-1β* in untreated or decitabine-treated Caco-2 cells (*p* = 0.035) and SK-CO-1 EVs injected cells (Figure 5E,F). 

However, when we investigated the association between the concentration of total HERV-positive EVs versus HERV-K or HERV-W positive EVs and *IL1-β* or *mpx*, we found that the concentration of HERV-K positive EVs was inversely associated with the immune response, as measured by *IL1-β* gene expression (HERV-K: β = −0.110; *p* = 0.013) and *mpx* (HERV-K: β = −0.110; *p* = 0.013) (Figure 6). 

## 3. Discussion

EVs are fundamental components of the metastatic niche and are involved in infiltration, metastasis, and immune tolerance processes during tumorigenesis. We hypothesized that HERV-positive EVs derived from tumor cell lines may be involved in tumor progression and may mediate immune tolerance. Therefore, we treated the two colorectal cancer cell lines Caco-2 and SK-CO-1, derived from non-metastatic epithelial and metastatic adenocarcinomas, respectively, with decitabine, to induce a global hypomethylation and to increase *HERV* expression. 

Although basal levels of LINE-1 methylation were similar in the Caco-2 and SK-CO-1 cells, *HERV* LTR methylation was significantly higher in Caco-2 than in SK-CO-1 untreated cells. It is well established that aberrant DNA methylation contributes to cancer development, and global hypomethylation is generally correlated with tumor grades. Furthermore, DNA hypermethylation at specific loci, in combination with repressive chromatin conformation, leads to the silencing of specific genes involved in tumorigenesis [34,35]. The Caco-2 cell line is representative of a more differentiated tumor grade, while SK-CO-1 is derived from a metastatic colorectal cancer characterized by lower methylation levels of *HERV* LTRs (especially *HERV-K* and *HERV-H* LTRs), as they are involved in tumor infiltration and metastasis [36]. As a matter of fact, *HERV-K* and *HERV-H env* expression levels were also generally higher in the SK-CO-1 cells than in the Caco-2 cells. 

Decitabine treatment induced reduction of *HERVs* methylation levels and increased *HERVs* gene expression. *HERVs* expression is often associated with cancer progression, and the expression of *HERV-H* has been previously observed in colorectal cancer [31,37]. Interestingly, transcriptional differences in *HERV-H* elements between colon cancer and adjacent normal tissues have been reported, thereby suggesting the active role that HERV-H plays in colorectal cancer tumorigenesis [30]. Moreover, *HERV-H* mRNA was reported to be elevated in metastatic tumor cells undergoing epithelial-to-mesenchymal transition [10]. *HERV-H* is not expressed in normal tissues, but is detected at high levels in cancer cells [18]. In particular, *HERV-H* transcripts were reported to be selectively transcribed in about 60% of colon cancers and in an even higher proportion of metastatic colon cancers [38]. Also, *HERV-K* is transcriptionally silent in normal cells, and becomes active after malignant transformation. Furthermore, increased expression of *HERV-K* has been detected in different human cancers, as well as HERV-P env, which was reported to be significantly overexpressed in lung and breast cancer patients in comparison to normal cases [26,27,39].

Although the Caco-2 and SK-CO-1 cells reacted similarly to decitabine treatment, by modulating *HERVs* methylation and expression, our experiments showed that the two cell lines do not display the same behavior in terms of EV release following decitabine treatment. The concentration of total EVs derived from the Caco-2 cells was higher than that of the SK-CO-1 cells in both untreated and decitabine-treated conditions. However, decitabine treatments determined an increase in EV release in the Caco-2 cells, but a decrease in the SK-CO-1 cells. A possible speculation is that the two cell lines are representative of two different cancer stages. While the Caco-2 cells are still partially differentiated cells, and therefore represent a more precocious step of tumorigenesis, in which immunosuppression is crucial for tumor survival, the SK-CO-1 cells have already reached the metastatic stage and might have developed alternative strategies to counterbalance an immune response.

We further characterized EVs in order to quantify EpCAM, HERV-K, and HERV-W as membrane determinants. EpCAM-positiveEVs were particularly high in the Caco-2 cells, confirming the differentiated condition of this type of colorectal adenocarcinoma-derived-cell line. On the contrary, in the SK-CO-1 cells, EpCAM-positive EV concentration was at low levels.

In order to investigate the effects of EVs derived from the Caco-2 and SK-CO-1 cells on innate immunity, we took advantage of the zebrafish model. Indeed, zebrafish embryos have been gaining favor as a powerful tool for the study of innate immunity [40]. In particular, the inflammatory response following innate immunity activation has been demonstrated to occur in zebrafish through the release of cytokines that are evolutionarily conserved in mammals [33,41]. The injection of EVs derived from both Caco-2 and SK-CO-1 cells in zebrafish embryos significantly lowered the expression of the pro-inflammatory cytokine *IL1-β* in comparison to embryos injected with control cell media, indicating a role of EVs in modulating innate immune response. Therefore, to specifically evaluate the net effect of HERV-positive EVs on the innate immune response in zebrafish, we considered the association between HERV-K positive EV concentrations and expression levels of *IL1-β* or *mpx*. Interestingly, we found a strong association between these two variables, thereby supporting our hypothesis that the HERV-positive EVs might act as immunomodulators in tumor progression. Indeed, in the presence of a robust block of pro-inflammation, EV injection generated an anti-inflammatory response, measured in terms of an increase in *IL-10* expression. 

Although our observations are associative in nature, and will require further experimental confirmations, the biological plausibility of our hypothesis is supported by HERV molecular features. In fact, HERVs include an immunosuppressive domain in their transmembrane envelope proteins, which may support tumor progression, abrogating the anti-oncogenic cytolytic immune response [42,43]. Moreover, the transmembrane subunit of a HERV-K *env* sequence was reported to inhibit T cell activation in a similar way to that of the HIV virus, influencing cytokine release and immune gene expression [44]. 

The present study must be interpreted taking into account both its strength and limitations. To the best of our knowledge, this was the first investigation describing the different steps from *HERV*s hypomethylation, to cellular expression, to release in EVs, and to functional effect on innate immune system. Therefore, our hypothesis is supported by different complementary approaches. Moreover, this is one of the first papers describing the use of zebrafish embryos as a suitable model to investigate the effects of EVs on conserved biological mechanisms.

However, we are aware that only HERVs and EpCAM determinants were evaluated on the surface of EVs, but the effects we observed on the immune response in zebrafish might be due to other components, as we injected the total amount of EVs derived from cells. Nonetheless, we observed a significant association between HERV-K positive EVs and a lower expression of *IL1-β* in zebrafish. The effect seems to be not generalized, because no association was detected for HERV-W positive EVs. Therefore, it is possible that HERV-W is not involved in this process, or at least not in the Caco-2 and SK-CO-1 colorectal cancer cells.

Moreover, decitabine treatment hypomethylates not only *HERVs* and repetitive elements (e.g., *LINE-1*), but also many other sequences. Therefore, it is possible that the effect of decitabine treatment might be due to other reactivating components. 

Future studies are needed to better characterize EVs derived from colorectal cancer cells also for the other HERV families (i.e., HERV-H and HERV-P) following specific antibodies generation. Moreover, a crucial perspective is to identify the mechanisms by which EVs, in particular those carrying HERVs, promote immune tolerance. In order to confirm the immunomodulatory role of HERV-positive EVs, in particular as regulators of innate immune response, the panel of investigated cytokines should be further implemented and the involvement of immune cells should be better defined. The characterization of the EV cargos will be pivotal in this context, as it will not only allow for the identification of the specific molecular targeted pathways, but it will also open new perspectives toward the modulation of immune response in cancer therapy.

## 4. Materials and Methods

### 4.1. Cell Cultures and Treatments

The human Caco-2 cell line (ATCC^®^ HTB-37, Basel, Switzerland) was derived from the colorectal adenocarcinoma of a 72-year old Caucasian male patient, while the SK-CO1 cell line (ATCC^®^ HTB-39, Basel, Switzerland) was derived from the metastatic site ascites, from a colorectal adenocarcinoma of a 65-year old Caucasian male patient. The Caco-2 cell line was grown and sub-cultured in DMEM medium (Sigma-Aldrich, St. Louis, MO, USA), while the SK-CO-1 cell line was grown and sub-cultured in EMEM medium (Sigma-Aldrich, Germany), both containing L-glutamine (Sigma-Aldrich), 10% *v*/*v* fetal bovine serum, and antibiotics (50 U/mL penicillin; 50 µg/mL streptomycin; Euroclone, Milan, Italy). All cells were maintained in an incubator at 37 °C in a 5% CO_2_ humidified atmosphere. Decitabine (5-Aza-2′-deoxycytidine, Sigma-Aldrich) treatments were performed at a final concentration of 5 µM, 96 h before harvesting the cells. For each condition, at least three independent experiments were conducted. After each experiment, cells were cultured in serum-free media for 24 h before medium collection and cell harvesting.

### 4.2. EVs Analysis

The isolation, purification, and characterization of EVs were performed by following the MISEV 2018 guidelines [45]. Detailed procedures and approaches are described in the Appendix A and Methods and in Appendix A. 

### 4.3. Isolation and Purification of EVs

For the isolation of EVs shed by cell culture, 8 mL of medium was collected from each flask and centrifuged at 1000, 2000, and 3000× *g* for 15 min at 4 °C. The obtained pellets were discarded to remove cell debris. EVs were then isolated from supernatants by ultracentrifugation at 110,000× *g* for 4 h at 4 °C in polypropylene ultracentrifuge tubes (Beckman Coulter; Brea, CA, USA), filled with PBS previously filtered through a 0.10 μm pore-size polyethersulfone filter (StericupRVP, Merck Millipore; Burlington, MA, USA). To carry out nanoparticles tracking analysis (NTA) and flow cytometry, the EV-rich pellet was resuspended in 500 µL of triple-filtered PBS.

### 4.4. Nanoparticle Tracking Analysis (NTA) of EVs

Numbers and dimensions of EVs were assessed by NTA, using the NanoSight NS300 system (Malvern Panalytical Ltd., Malvern, UK), as previously described [46], which measures the Brownian motion of particles suspended in fluid and displays them in real time through a high sensitivity CCD camera. Five 30 s recordings were made for each sample. Collected data were analyzed with NTA software (Malvern Panalytical Ltd.), which provided high-resolution particle-size distribution profiles, as well as measurements of the EV concentration.

### 4.5. High Resolution Flow Cytometry on EVs

EVs were characterized by high-resolution flow cytometry (MACSQuant, Miltenyi Biotec, Bergisch Gladbach, Germany), according to the protocol for EV characterizations detailed in the Appendix A and Methods for high-resolution flow cytometry. Briefly, sample acquisition was performed at the minimum speed flow (25 µL/min) using a MACSQuant Analyzer (Miltemyi Biotec). Sheath fluid was filtered through 0.1 μm pore size filter to further improve the signal-to-noise ratio. The fluorescent beads Fluoresbrite^®^ YG Carboxylate Microspheres Size Range Kit I (0.1, 0.2, 0.5, 0.75, and 1 μm) (Polysciences Inc, Warrington, Pennsylvania) was used to set the calibration gate in the FSC/FL1 and FSC/SSC dot plots. Using a side scatter (SSC) threshold of 10 arbitrary units, the lower sensitivity of the instrument was determined and the SSC and FITC voltages were set up. An overlap in the 100 nm beads population and the background noise was observed. In this way, it was possible to gate the MVs ≥ 200 nm diameter. A total of 30 μL of sample was acquired on the MACSQuant Analyzer. Event numbers, analyzed at a low flow rate and below 10,000 events/second, of equal sample volumes were counted. Information about concentration (N^o^. events/μL) was calculated by the analyzer software. To verify the correctness of the count, we performed a serial dilutions measurement of different EV samples. An r-value higher than 0.9 demonstrated the goodness of the experiment set up. From these data we set the resuspending volume of 500 µL of PBS, resulting from the midpoint of the standard curve. 

To analyze cell culture media-isolated EV integrity, 60 µL aliquots were stained with 0.2 μM 5(6)-carboxyfluorescein diacetate N-succinimidyl ester (CFSE) at 37 °C for 20 min in the dark. In order to assess the cellular origin of the EVs isolated from plasma, an immunophenotypization assay was performed for each sample using a panel of specific antibodies: anti-HERV-K (clone 5i73) (USBiological, Salem, MA, USA), anti-HERV-W (clone clone 4F10) (Sigma Aldrich), anti-CD326 (EpCAM)-APC (clone HEA-125) (Miltenyi Biotec), anti-CD63 (clone-REA1055) (Miltenyi Biotec). Each antibody aliquot was previously centrifuged at 17,000× *g* for 30 min at 4 °C to eliminate aggregates. A stained PBS control sample was used to detect the autofluorescence of each antibody. Quantitative multiparameter analysis of flow cytometry data was carried out using FlowJo Software (Tree Star, Inc.; Ashland, OR, USA). Sample plots and gating for each antibody are shown in the Appendix A for negative control (i.e., triple-filtered PBS), Caco-2, and SK-CO-1 cells.

### 4.6. Methylation Analysis

DNA was isolated from 1 × 10^6^ cells, using the QIamp DNA Mini Kit (Qiagen, Hilden, Germany), following the manufacturer’s protocol. The analysis of DNA methylation was performed by following previously published methods [47], with minor modifications. Briefly, a 50 μL PCR reaction was carried out with 25 μL of Hot Start GoTaq Green Master mix (Promega, Madison, WI, USA), 1 pmol of forward primer, 1 pmol of reverse primer, and 25 ng of bisulfite-treated genomic DNA. PCR cycling conditions and primer sequences are provided in Appendix A. Biotin-labeled primers (forward or reverse depending on the assay) were used to purify the final PCR product with Sepharose beads. The PCR product was bound to Streptavidin Sepharose beads (Amersham Biosciences, Uppsala, Sweden), purified, washed, denatured with 0.2 M NaOH, and washed again with the Pyrosequencing Vacuum Prep Tool (Pyrosequencing, Inc., Westborough, MA, USA), according to the manufacturer’s instructions. Pyrosequencing primer (0.3 μΜ) was annealed to the purified single-stranded PCR products, and pyrosequencing was performed with the PyroMark MD System (Pyrosequencing, Inc., Westborough, MA, USA). Methylation levels were expressed as the percentage of cytosines that were methylated, determined as the number of methylated cytosines divided by the sum of methylated and unmethylated cytosines, multiplied by 100% (% 5-methyl-Cytosine). 

### 4.7. RNA Extraction form Caco-2 and SK-CO-1 Cells, Reverse Transcription-PCR, and Real-Time Quantitative-PCR Assays (RT-qPCR)

RNA was isolated from 1 × 10^6^ cells using the QIAmp RNA Blood Mini Kit (Qiagen, Hilden, Germany), following the manufacturer’s protocol. One microgram of the total RNA was subjected to reverse transcription using the QuantiTec Reverse Transcription Kit (Qiagen, Hilden, Germany). Relative quantitative real-time PCR was performed to evaluate the expression levels of the HERV-H, -K, -P, and -W *env* genes in a 7500 real-time PCR system (Applied Biosystem, CA, USA) using the QuantiTect SYBR Green PCR kit (Qiagen, Hilden, Germany), according to the manufacturer’s instructions, with the primer sets shown in Appendix A. Forty cycles were performed for each real-time PCR assay, with the temperatures of annealing at 54 °C. All the samples were tested in triplicate and were compared to the mean expression of the housekeeping genes glyceraldehyde 3-phosphate dehydrogenase gene (GAPDH) and β-actin. Non-template controls were included for each primer pair. The quantification of the RNA expression was performed using the comparative Ct method, and the differences between the levels of *env* gene expression in the biological samples were calculated by relative quantification (RQ). 

### 4.8. Animals

Zebrafish (*Danio rerio*) embryos were raised and maintained according to international (EU Directive 2010/63/EU) and national guidelines (Italian decree 4 March 2014, n.26) on the protection of animals used for scientific purposes. The fish were maintained under standard conditions in the fish facilities of Bioscience Dept, University of Milan, Via Celoria 26-20133 Milan, Italy (Aut. Prot, n. 295/2012-A-20/12/2012). We expressed the embryonic ages in days post fertilization (dpf). Embryos were collected by natural spawning, staged according to Kimmel and colleagues [48], and raised at 28 °C in fish water (Instant Ocean, 0.1% Methylene Blue) in Petri dishes, according to established techniques. After 1 dpf, to prevent pigmentation 0.003% 1-phenyl-2-thiourea (PTU, Sigma-Aldrich, Saint Louis, MO, USA) was added to the fish water. Embryos were washed, dechorionated, and anaesthetized, with 0.016% tricaine (Ethyl 3-aminobenzoate methanesulfonate salt; Sigma-Aldrich), before EV injection.

### 4.9. EVs Microinjections

Zebrafish embryos at 2 dpf were manually dechorionated and anaesthetized with 0.016% tricaine (Ethyl 3-aminobenzoate methanesulfonate salt; Sigma-Aldrich). EVs derived from the Caco-2 and SK-CO-1 lines were resuspended in PBS 1X and injected into the duct of Cuvier, to allow their delivery in the circulation. A manual microinjector (Eppendorf, Germany) was used with glass microinjection needles. Control embryos were injected with serum-free media (DMEM for Caco-2 and EMEM for SK-CO1 cells). Following the injections, embryos were kept at 28 °C for 20 h, until their processing for RNA extraction. 

### 4.10. RNA Extraction form Zebrafish Embryos, Reverse Transcription-PCR, and RT-qPCR

Zebrafish embryos at 3 dpf (at least 30 embryos for each category), were collected for RNA extraction using TRIZOL reagents (Life Technologies, Carlsbad, CA, USA) following the manufacturer’s protocol. After DNase I RNase-free (Roche Diagnostics) treatment to avoid possible genomic contamination, 1 μg of RNA was reverse-transcribed using the “ImProm-II™ Reverse Transcription System” (Promega, Madison, WI, USA) and a mixture of oligo(dT) and random primers, according to manufacturer’s instructions. RT-qPCRs were carried out in a total volume of 20 μL containing 1X iQ SYBR Green Super Mix (Promega), using the proper amount of the RT reaction [49]. RT-qPCRs were performed using the BioRad iCycler iQ Real-Time Detection System (BioRad, Hercules, CA, USA). The expression levels of *IL1-β* were determined. qPCRs were performed using the BioRad iCycler iQ Real-Time Detection System (BioRad, Hercules, CA, USA). Thermocycling conditions were 95 °C for 10 min, 95 °C for 10 s, and 55 °C for 30 s. All reactions were performed in triplicate for 40 cycles. For normalization purposes, *rpl8* expression levels were tested in parallel with the gene of interest [50]. Primer sequences are reported in Appendix A.

### 4.11. Statistical Analysis

Descriptive statistics were performed on all variables: continuous variables were expressed as the mean ± standard deviation (SD) and the median with interquartile range (IQR). Spaghetti and box plots were used to represent *HERV* LTR and *LINE-1* methylation, *HERVs* expression levels, counts of EVs, and of EVs subtypes, of untreated and decitabine-treated Caco-2 and SK-CO-1 cells. Spaghetti and box plots were also used to represent experiments for *IL1-β* expression levels in zebrafish embryos injected with control medium, with EVs derived from decitabine-treated and with cell lines (in both Caco-2 and SK-CO-1).

Comparisons of *HERV* LTR methylation between the two cell types in untreated cells were evaluated by t-test and validated by the Wilcoxon rank-sum test. The same approach was adopted to compare untreated- and decitabine-treated cells in each cell line. 

We analyzed the effect of decitabine treatments in terms of distribution of vesicle mean concentrations for each EV size. Firstly, we estimated the EV mean concentration at each size, with univariate Poisson linear regression models for repeated measures. Secondly, we compared the EV mean differences between untreated and decitabine-treated cell lines. Results were reported as a series graph for EV mean concentrations of each size and vertical bar charts to represent the size-specific *p*-values obtained, comparing untreated- and decitabine-treated Caco-2 and SK-CO1 cells. For all the graphs the X-axis was the size of EVs. 

The association between the number of HERV-K and HERV-W positive EVs and *IL1-β* expression levels in zebrafish embryos was investigated with linear regression models adjusted for cell lines and treatment. All tests were two-sided, and a *p*-value < 0.05 was considered statistically significant. Analyses were performed with SAS 9.4 software (SAS Institute Inc., Cary, NC, USA).

## Figures and Tables

**Figure 1 ijms-20-03669-f001:**
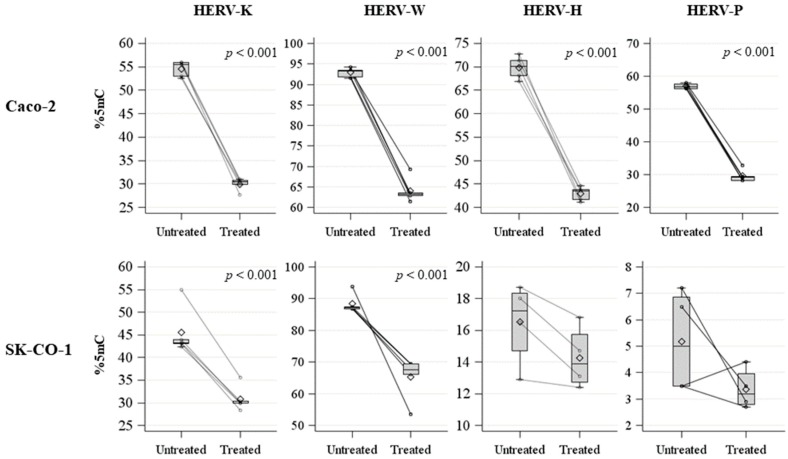
Spaghetti plot and box plot of *HERV-H*, *HERV-K*, *HERV-W*, and *HERV-P* LTRs methylation levels (%5mC) in untreated and treated Caco-2 and SK-CO-1 cells.

**Figure 2 ijms-20-03669-f002:**
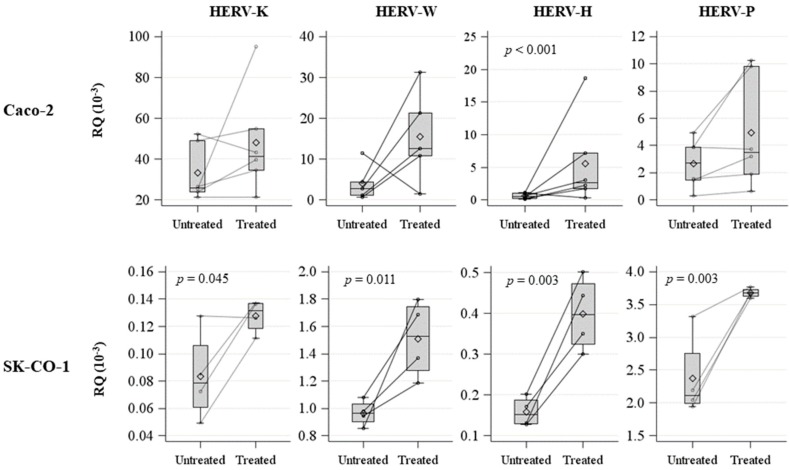
Spaghetti plot and box plot of *HERV-H*, *HERV-K*, *HERV-W*, and *HERV-P* expression levels (RQ 10^−3^) in untreated and decitabine-treated Caco-2 and SK-CO1 cells.

**Figure 3 ijms-20-03669-f003:**
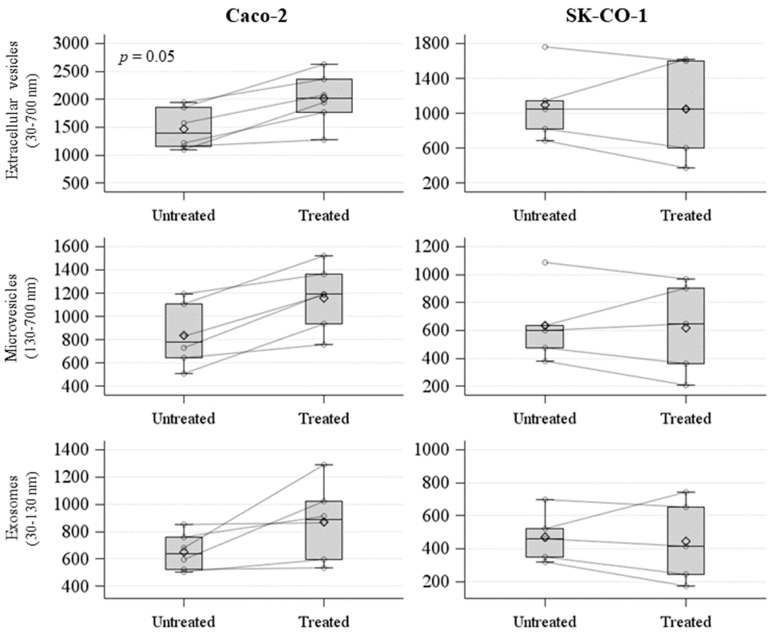
Spaghetti plot and box plot of number of extracellular vesicles (EVs) (count/mL × 10^6^) derived from untreated and decitabine-treated Caco-2 and SK-CO-1 cells.

**Figure 4 ijms-20-03669-f004:**
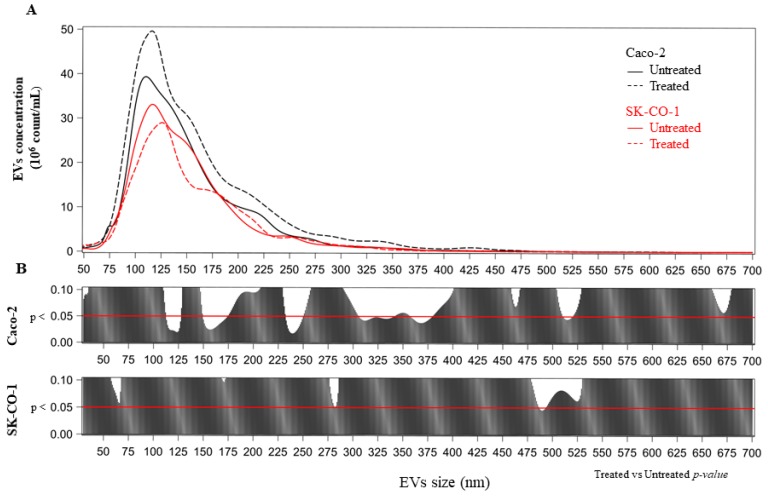
(**A**) Number of EVs (× 10^6^) for each size (nm) from Caco-2 and SK-CO-1 cells. (**B**) For each size, the *p*-value was reported from poisson regression models with repeated measures.

**Figure 5 ijms-20-03669-f005:**
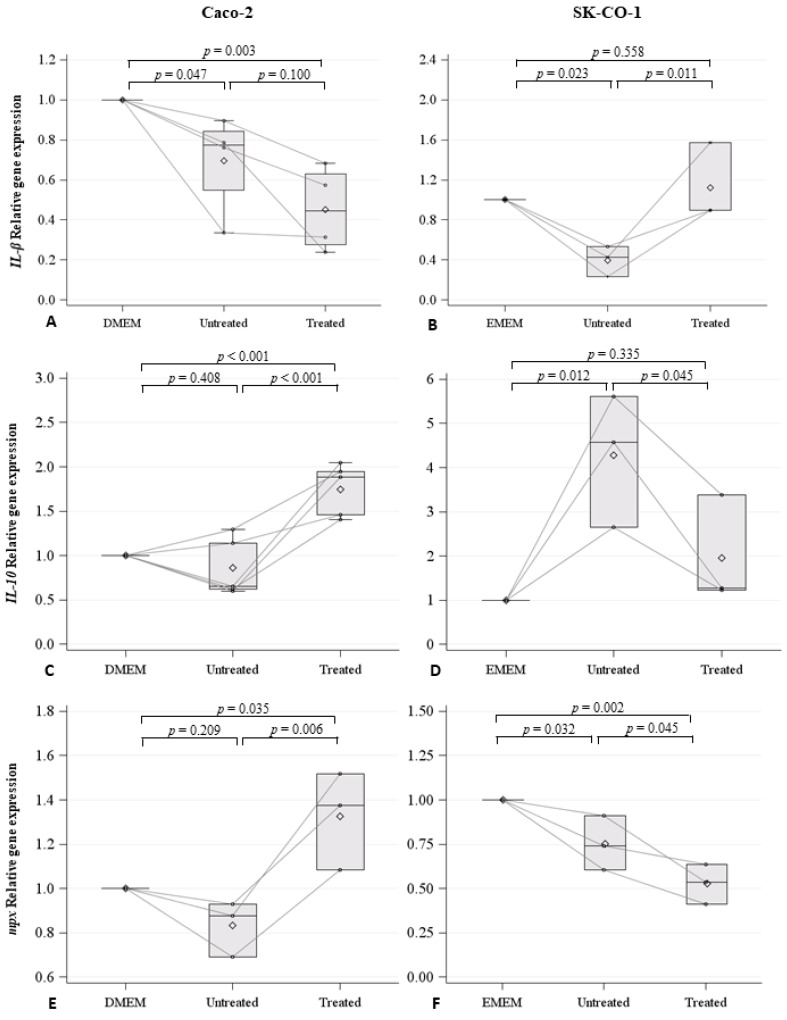
Spaghetti plot and box plot of *IL1-β* (**A**,**B**), *IL-10* (**C**,**D**), and *mpx* (**E**,**F**) expression levels (RQ) in zebrafish embryos injected with control medium (i.e., Dulbecco’s Modified Eagle’s Medium (DMEM) and Eagle’s Minimum Essential Medium (EMEM), respectively), with EVs derived from untreated Caco-2, decitabine treated Caco-2 cells, untreated SK-CO-1, and decitabine-treated SK-CO-1 cells.

**Figure 6 ijms-20-03669-f006:**
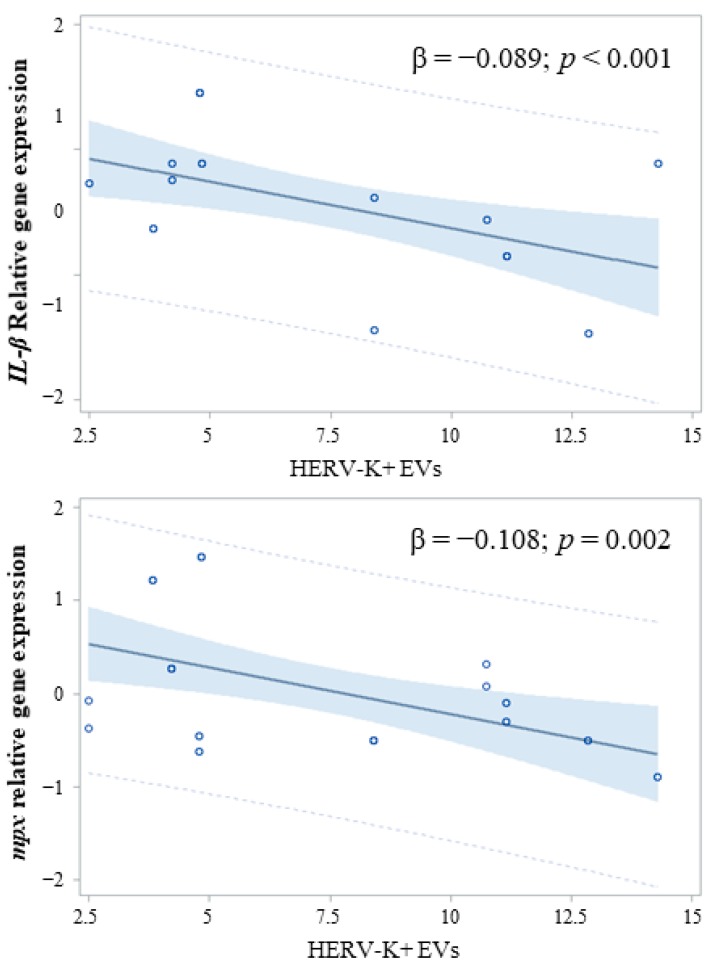
Association between HERV-K positive EVs and the expression of *IL1-β* and *mpx*.

**Table 1 ijms-20-03669-t001:** Association of *HERV* LTRs methylation between untreated Caco-2 and SK-CO-1 cell lines. T-test and Wilcoxon *p*-values in bold are statistically significant.

Element	Cell Type	Mean	Standard Deviation	Median	Interquartile Range	T-Test	Wilcoxon
HERV K	Caco-2	54.6	1.6	55.5	2.9	**0.016**	**0.036**
	SK-CO-1	45.5	5.3	43.3	0.8		
HERV W	Caco-2	92.9	1.1	93.2	1.6	**0.014**	0.095
	SK-CO-1	88.4	3.0	87.1	0.7		
HERV H	Caco-2	69.9	2.4	70.1	3.3	**<0.001**	**0.020**
	SK-CO-1	16.5	2.6	17.3	3.7		
HERV P	Caco-2	57.0	0.8	93.2	1.6	**<0.001**	**0.019**
	SK-CO-1	5.2	2.0	87.1	0.7		
LINE1	Caco-2	56.9	2.3	56.6	1.0	**0.002**	**0.012**
	SK-CO-1	51.5	1.6	50.8	0.8		

**Table 2 ijms-20-03669-t002:** Association of *HERV* LTRs methylation between untreated- and decitabine-treated- Caco-2 and SK-CO-1 cells. T-test and Wilcoxon *p*-values in bold are statistically significant.

**Caco-2 Cells**							
**Element**	**Treatment**	**Mean**	**Standard Deviation**	**Median**	**Interquartile Range**	**T-Test**	**Wilcoxon**
HERV-K	Untreated	54.6	1.6	55.5	2.9	**<0.001**	**0.012**
	Treated	30.0	1.3	30.5	0.9		
HERV-W	Untreated	92.9	1.1	93.2	1.6	**<0.001**	**0.012**
	Treated	64.1	3.0	63.2	63.2		
HERV-H	Untreated	69.9	2.4	70.1	3.3	**<0.001**	**0.012**
	Treated	42.9	1.5	43.5	2.1		
HERV-P	Untreated	57.0	0.8	56.8	1.3	**<0.001**	**0.012**
	Treated	29.6	1.9	29.2	1.3		
**SK-CO-1 Cells**							
**Element**	**Treatment**	**Mean**	**Standard Deviation**	**Median**	**Interquartile Range**	**T-Test**	**Wilcoxon**
HERV-K	Untreated	45.5	5.3	43.3	0.8	**<0.001**	**0.012**
	Treated	30.8	2.8	29.9	0.4		
HERV-W	Untreated	88.4	3.0	87.1	0.7	**<0.001**	**0.012**
	Treated	65.2	6.7	67.5	3.3		
HERV-H	Untreated	16.5	2.6	17.3	3.7	0.210	0.312
	Treated	14.3	2.0	13.9	3.0		
HERV-P	Untreated	5.2	2.0	5.0	3.4	0.137	0.183
	Treated	3.4	0.8	3.2	1.2		

**Table 3 ijms-20-03669-t003:** Characterization of EV concentrations (count/mL) in Caco-2 an SK-CO1 cells.

Treatment		Caco-2		SK-CO-1	
		Mean	Standard Deviation	Mean	Standard Deviation
Untreated	HERV-K	5.5	3.9	10.9	4.1
	HERV-W	9	3.4	13.3	4.5
	EpCAM	145.5	132.1	119.7	105.1
	CD63	131.7	173.2	34.7	25.1
Treated	HERV-K	9.1	6	8.3	4.1
	HERV-W	10.5	3.6	10.8	4.6
	EpCAM	204.6	184.2	81.4	58.5
	CD63	133.4	190.4	58.2	59.7

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
