# Peer review of "Extracellular Vesicles Released by Colorectal Cancer Cell Lines Modulate Innate Immune Response in Zebrafish Model: The Possible Role of Human Endogenous Retroviruses"

_ijms, 2019, doi:10.3390/ijms20153669_

Round 1
Reviewer 1 Report
Authors have addressed my comments by performing cytokine experiment and by evaluating retroviral contamination in EVs pellet. My opinion is ''accept in present form''.
Reviewer 2 Report
Most of my concerns were addressed by authors now. Thus, I would recommend this manuscript for the publication in IJMS.
This manuscript is a resubmission of an earlier submission. The following is a list of the peer review reports and author responses from that submission.
Round 1
Reviewer 1 Report
Authors have selected a topic of scope, however the study design leaves behind some gap.
1. Major concern is the claim about immune modulation by EVs. "EVs modulate innate immune response" is a big claim unless immunomodulatory roles of EVs satisfy the requirements by analyzing major players of immune modulation. Measuring IL1-β only, is not enough to make the claim. Authors should measure the expression of additional cytokines, and the involvement of immune cells. i.e. additional experiments are required.
Other concerns
2. Title should refer to what has been reported (E.g. modulate innate immune response in Zebrafish model), so that readers should not be misguided by title.
3. How authors correlate Zebrafish model with tumour microenvironment or any diseases model of interest.
4. While claiming EVs are positive for retrovirus, authors cannot exclude the possibility of co-pelleting/aggregation of retrovirus with EVs isolated by ultracentrifugation. Since retroviruses share the same size of EVs it is likely that retroviruses are pelleted at shared size and density. Authors need to validate whether retrovirus are packed inside EVs and have not just pelleted with EVs as aggregates. Additional experiment is required.
5. Finally, authors should consider MISEV-2018 guidelines for minimal experimental requirements (PMID: 30637094).
Author Response
1. We thank the reviewer 1 for the timely comment, and according to his/her request, we analyzed further players of immune modulation in zebrafish: first the anti-inflammatory cytochine Interleukin 10 (IL-10) and the myeloperoxidase (mpx) marker of activated neutrophils.
The results are reported in the new Figures 5 and 6 and are described in the draft section “Effects on innate immune response in zebrafish embryos”.
Moreover, we added the following paragraph in Discussion: “In order to confirm the immunomodulatory role of HERV-positive EVs, in particular as regulators of innate immune response, the panel of investigated cytokines should be further implemented and the involvement of immune cells should be better defined.”
2. We agree with the reviewer and changed the title accordingly: “EXTRACELLULAR VESICLES RELEASED BY COLORECTAL CANCER CELL LINES MODULATE INNATE IMMUNE RESPONSE IN ZEBRAFISH MODEL: THE POSSIBLE ROLE OF HUMAN ENDOGENOUS RETROVIRUSES”.
3. We injected tumor-cell derived EVs in zebrafish embryos in order to investigate their effects on innate immune response. Indeed, metastatic niche, which is constituted by a complex microenvironment, favors the escape of the surveillance by the immune system, thus promoting tumor proliferation. We chose the zebrafish model as zebrafish embryos have only the innate immunity until the first month of development, allowing to specifically observe the net effect on this immune component. In particular, the inflammatory response following innate immunity activation has been demonstrated to occur in zebrafish through the release of cytokines that are evolutionarily conserved from zebrafish to human. This condition makes the zebrafish a suitable model to investigate the effects of human EVs on the innate immunity. Therefore, we strongly believe that the use of this tool provides crucial information on how tumor-derived EVs are able to influence innate immune response.
To better clarify this point, we added the following sentence into the section “Discussion”:
“In order to investigate the effects of EVs derived from the Caco-2 and SK-CO-1 cells on innate immunity, we took advantage of the zebrafish model. Indeed, zebrafish embryos have been gaining favor as a powerful tool for the study of innate immunity. In particular, the inflammatory response following innate immunity activation has been demonstrated to occur in zebrafish through the release of cytokines that are evolutionarily conserved in mammals [33,40]”.
4. The flow-cytometry analysis allowed us to measure the intact EVs by using the 5(6)-Carboxyfluorescein diacetate N-hydroxysuccinimidyl ester (CFSE), a cell permeant, non-fluorescent pro-dye. If incorporated into intact EVs, which contain esterases as live cells, the acetate groups of CFSE is cleaved producing a membrane-impermeant molecule with green fluorescence. This tool allows to selectively gate intact EVs, on which the surface-expression of specific proteins is evaluated (e.g. CD63, EpCAM, HERV-K, HERV-W) as reported in the new Supplementary Figures S5-S9. In order to evaluate whether also restroviruses were pelleted together with EVs, we measured HERV-W and HERVK positive events in samples without gating for CFSE, and observed that HERV-K and HERV-W concentration was the same in both cases (i.e. gating or not for CFSE), as reported in the new Supplementary Figures S3 (Caco-2) and S4 (SK-CO-1). Considering the subpopulations of EVs analyzed by high-resolution flow cytometry, these results minimize the possibility of pelleting HERV-W and HERV-K positive particles with dimensions ≥ 100 nm. Nonetheless, we cannot exclude the presence of HERVs free proteins (with dimension ≤ 100 nm) in the samples, due to the intrinsic limits of High Resolution Flow Cytometry (please, see also Supplementary Material and Methods).
We added the following sentence to results section “Evaluation and Characterization of EVs”:
“Since retroviruses share the same size of EVs, we evaluated the possibility of having a retroviral contamination in EV pellet. Therefore, we measured HERV-W and HERV-K positive events in samples without gating for CFSE staining, and observed that HERV-K and HERV-W concentrations were the same in both cases (i.e. either gating or not for CFSE), as reported in the Supplementary Figure S2 for Caco-2 cells, and Supplementary Figure S3 for SK-CO-1 cells. Considering the subpopulations of EVs analyzed by high-resolution flow cytometry, these results minimize the possibility of pelleting HERV-W and HERV-K positive particles with dimensions ≥ 100 nm.”
5. We thank the reviewer for the comment. As we followed MISEV-2018 guidelines in performing our experiments, we further detailed the section “Material and Methods” and added Supplementary Materials and Methods where we widely illustrated the methods used for isolation and analysis of EVs (as required by MISEV-2018 guidelines).
We added the section “EVs analysis in Material and Methods”:
“EVs analysis
Isolation, purification, and characterization of EVs were performed by following MISEV 2018 guidelines [43]. Detailed procedures and approaches are described in Supplementary Materials and Methods and in Supplementary Figures S5-S9.”
Reviewer 2 Report
The manuscript by Ferrari et al. described that colon cancer-derived EVs regulate innate immune response by regulating IL1b expression in the zebrafish embryo. The authors found that decitabine treatment increases the amount of EVs in colon cancer cells and also increase HERV expressions. In addition, the authors showed that HERV-K EV amount is negatively correlated with IL1-B.
Although it is important that cancer EVs modulate innate immunity in vivo, there are many concerns on the experimental designs as shown below. Thus, at least in the present form, this manuscript is not acceptable in IJMS.
Major comments
1. For in vivo zebrafish embryo experiment, the authors collected EVs from colon cancer cell lines. According to their materials and methods, they cultured cancer cell using medium with 10% FBS. Basically, FBS contains a lot of bovine EVs. To test EV function, it is necessary to use serum-free condition or EVs-depleted FES. Otherwise, it is very hard to accurately test the function of cancer cell-derived EVs.
2. Based on the FACS analysis, the authors characterized HERV-K or HERV-W positive EVs. It is not clear if this analysis accurately quantifies the EVs amount. Is HERV-K or HERV-W located on the EVs? If they are located in the EVs, how does this experiment works? The authors should explain this point.
3. After collecting the EVs with/without decitabine treatment, the authors should perform qPCR to quantify the amount of each HERV.
4. In zebrafish embryo experiment, the authors assessed only IL1-b expression. It is better to assess a series of cytokine expressions that are related to immune response.
5. In Fig 5 (SK-CO-1), decitabine-treated EVs from SK-CO-1 significantly induced IL1-B. It is totally opposite data, as compared to Caco-2 and untreated SK-CO-1. The authors should discuss more the reason and the possibility of these results.
6. For the characterization of EVs, the authors should perform western blot for EV markers such as CD63, CD81, CD9, and Alix.
Minor comment
1. In Fig 3, the author mentioned a significant increase of total EVs in Caco-2 cells after treatment. But, based on the graph, it is “p = 0.052”. It is not < 0.05. Please change the description.
Author Response
Major comments
1. we are extremely thankful to the reviewer as his/her observation allowed us to uncover an error in the method section, which made us erroneously report that complete medium was used to grow cells and isolate EVs, while EV isolation was performed after 24 h of starvation. We have included this information in the methods, and described in detail the entire protocol in Supplementary methods (MISEV minimal requirements). We apologize for the mistake.
2. As replied to Reviewer#1 point 4, the flow-cytometry analysis allowed us to measure the intact EVs by using the 5(6)-Carboxyfluorescein diacetate N-hydroxysuccinimidyl ester (CFSE), a cell permeant, non-fluorescent pro-dye. If incorporated into intact EVs, which contain esterases as live cells, the acetate groups of CFSE is cleaved producing a membrane-impermeant molecule with green fluorescence. This tool allows to selectively gate intact EVs, on which the surface-expression of specific proteins is evaluated (e.g. CD63, EpCAM, HERV-K, HERV-W) as reported in the new Supplementary Figures S5-S9. In order to evaluate whether also restroviruses were pelleted together with EVs, we measured HERV-W and HERVK positive events in samples without gating for CFSE, and observed that HERV-K and HERV-W concentration was the same in both cases (i.e. gating or not for CFSE), as reported in the new Supplementary Figures S3 (Caco-2) and S4 (SK-CO-1). Considering the subpopulations of EVs analyzed by high-resolution flow cytometry, these results minimize the possibility of pelleting HERV-W and HERV-K positive particles with dimensions ≥ 100 nm. Nonetheless, we cannot exclude the presence of HERVs free proteins (with dimension ≤ 100 nm) in the samples, due to the intrinsic limits of High Resolution Flow Cytometry (please, see also Supplementary Material and Methods).
We added the following sentence to results section “Evaluation and Characterization of EVs”:
“Since retroviruses share the same size of EVs, we evaluated the possibility of having a retroviral contamination in EV pellet. Therefore, we measured HERV-W and HERV-K positive events in samples without gating for CFSE staining, and observed that HERV-K and HERV-W concentrations were the same in both cases (i.e. either gating or not for CFSE), as reported in the Supplementary Figure S2 for Caco-2 cells, and Supplementary Figure S3 for SK-CO-1 cells. Considering the subpopulations of EVs analyzed by high-resolution flow cytometry, these results minimize the possibility of pelleting HERV-W and HERV-K positive particles with dimensions ≥ 100 nm.”
3. Following collection of EVs, we extracted total RNA and quantified HERV transcript expression in cells by qPCR in both decitabine-treated and untreated conditions. The experiments are detailed in the section “HERV methylation analysis” in the Results (see also Figure 2 and Supplementary Table S1). We did not evaluate HERV transcript levels in cells-derived EVs, as we hypothesized that the functional role we are investigating (i.e. targeting the immune cells) may be mediated by HERV proteins on EV surfaces. We therefore evaluated HERV proteins expression on EVs as reported in “Evaluation and Characterization of EVs” section in the Results.
4. We thank the reviewer for the timely comment, and according to his/her request, we analyzed further players of immune modulation in zebrafish: first the anti-inflammatory cytochine Interleukin 10 (IL-10) and the myeloperoxidase (mpx) marker of activated neutrophils.
The results are reported in the new Figure 5 and 6 and are described in the draft section “Effects on innate immune response in zebrafish embryos”.
5. IL1-B expression increased following treatment with decitabine in SK-CO-1 cells, while it decreased in Caco-2 cells. However, the mean of total EVs concentrations were reduced in SK-CO-1 treated cells compared with those of untreated, and the mean concentrations were the lowest for almost every dimension (Figure 2A). On the contrary, EVs concentrations of Caco-2 cells treated with decitabine increased in comparison with those of untreated and were the highest for all the dimensions (Figure 2A). Moreover, the evaluation of HERV-K positive EVs concentration by high-resolution flow cytometry showed a tendency of decrease (although not significant) between untreated and decitabine-treated SK-CO-1 cells, while in Caco-2 we observed a tendency of increase, as reported in Supplementary Figure 2. Given this evidence, we hypothesized that HERV-K expression on EVs might be associated with IL1-B expression in zebrafish following the injection of tumor-derived EVs. To confirm our hypothesis, we investigated this association by a linear regression model adjusting for cell lines and treatments. Interestingly we observed that the concentration of HERV-K positive EVs was inversely associated to the immune response, as measured by IL1-β gene expression (HERV-K: β=-0.110; p=0.013), as reported in Figure 6. We think that these results support our hypothesis. HERV-K positive EVs might act as immunomodulators in tumor progression, even though future functional analysis are needed to confirm this hypothesis.
6. We evaluated CD63 expression on EVs by flow-cytometry, and reported the results Table 3. We added the following sentence in the section Results “Evaluation and Characterization of EVs”:
“In addition, we quantified the tetraspanin CD63 positive EVs as a non-tissue specific marker for EVs (Table 3)”.
The characterization of EVs was performed by using High Resolution Flow Cytometry, as indicated by MISEV 2018 guidelines. We chose this approach as it allows evaluating the amount of intact EVs. To do this, we treated EVs with the 5(6)-Carboxyfluorescein diacetate N-hydroxysuccinimidyl ester (CFSE), a cell permeant, non-fluorescent pro-dye. If incorporated into intact EVs, which contain esterases as live cells, the acetate groups of CFSE is cleaved producing a membrane-impermeant molecule with green fluorescence. This tool allows to selectively gate intact EVs, on which the surface-expression of specific proteins is evaluated (e.g. CD63, EpCAM, HERV-K, HERV-W). We integrated the section “Material and Methods” and further described the procedures we applied in this study.
We chose this approach as Western Blot analysis, which is widely used in EVs study, evaluates the expression of proteins, without discriminating between cell membrane fragments, intact, or broken EVs. The evaluation of intact EVs is an indispensable condition for the functional studies we performed in the zebrafish model, by which we investigated the potential functional role of intact Caco-2- and SK-CO-1- derived EVs.
To better explain our approach, we added supplementary materials and methods, Supplementary Figures S2, S3 and S5-S9, and integrated the section “High resolution flow cytometry on EVs” as follows:
“EVs were characterized by High Resolution Flow Cytometry (MACSQuant, Miltenyi Biotec, Germany) according to the protocol for EV characterizations detailed in supplementary materials and methods. Briefly, samples acquisition was performed at the minimum speed flow (25 μl/min) using a MACSQuant Analyzer (Miltemyi Biotec). Sheath fluid was filtered through 0.1μm pore size filter to further improve the signal-to-noise ratio. The fluorescent beads Fluoresbrite® YG Carboxylate Microspheres Size Range Kit I (0,1, 0.2, 0.5, 0.75, and 1μm) (Polysciences Inc, Warrington, Pennsylvania) were used to set the calibration gate in the FSC/FL1 and FSC/SSC dot plots. Using a side scatter (SSC) threshold of 10 arbitrary units the lower sensitivity of the instrument was determined and the SSC and FITC voltages were set up. An overlap in the 100nm beads population and the background noise was observed. In this way, it was possible to gate the MVs ≥ 200 nm diameter. 30 μL of sample was acquired on the MACSQuant Analyzer. Event numbers, analyzed at low flow rate and below 10,000 events/second, of equal sample volumes were counted. Information about concentration (No. events/μL) were calculated by the analyzer software. To verify the correctness of the count, we performed a serial dilutions measurement of different EV samples. r-value higher than 0,9 demontrated the goodness of the experiment set up. From this data we set the resuspending-volume of 500 μl of PBS resulted from the midpoint of the standard curve.
To analyze cell culture media-isolated EV integrity, 60 μl aliquots were stained with 0.2 μM 5(6)-carboxyfluorescein diacetate N-succinimidyl ester (CFSE) at 37 °C for 20 min in the dark. In order to assess the cellular origin of the EVs isolated from plasma, an immunophenotypization assay was performed for each sample using a panel of specific antibodies: anti-HERV-K (clone 5i73) (USBiological, MA, USA), anti-HERV-W (clone clone 4F10) (Sigma Aldrich), anti-CD326 (EpCAM)-APC (clone HEA-125) (Miltenyi Biotec), anti-CD63 (clone-REA1055) (Miltenyi Biotec). Each antibody aliquot was previously centrifuged at 17,000 × g for 30 min at 4 °C to eliminate aggregates. A stained PBS control sample was used to detect the autofluorescence of each antibody. Quantitative multiparameter analysis of flow cytometry data was carried out by using FlowJo Software (Tree Star, Inc.; Ashland, OR, USA). Sample plots and gating for each antibody are shown in the Supplementary Figures S5-S9 for negative control (i.e. triple filtered PBS), Caco-2, and SK-CO-1 cells.”
minor comments
We modified the Figure 3 as suggested by the Reviewer 2 (p=0.05), and changed the description in the manuscript as follows:
“We observed an increase of total EVs release in the decitabine treated Caco-2 cell line, with an average of 2.01*109/mL (±4.71*109/mL), compared to the untreated, with an average of 1.47*109/mL (±3.69*109/mL) (p=0.05) (Figure 3).”